# Enhancer of Zeste Homolog 2 (EZH2) Contributes to Rod Photoreceptor Death Process in Several Forms of Retinal Degeneration and Its Activity Can Serve as a Biomarker for Therapy Efficacy

**DOI:** 10.3390/ijms22179331

**Published:** 2021-08-28

**Authors:** Martial Mbefo, Adeline Berger, Karine Schouwey, Xavier Gérard, Corinne Kostic, Avigail Beryozkin, Dror Sharon, Hélène Dolfuss, Francis Munier, Hoai Viet Tran, Maarten van Lohuizen, William A. Beltran, Yvan Arsenijevic

**Affiliations:** 1Unit of Retinal Degeneration and Regeneration, Department of Ophthalmology, University of Lausanne, Fondation Asile des Aveugles, 1004 Lausanne, Switzerland; mmbefo@gmail.com (M.M.); adeline.berger@fa2.ch (A.B.); karine.schouwey@gmail.com (K.S.); xaviergerard26@yahoo.fr (X.G.); corinne.kostic@fa2.ch (C.K.); 2Hadassah Medical Center, Faculty of Medicine, The Hebrew University Jerusalem, Jerusalem 91120, Israel; avigail.beryozkin@mail.huji.ac.il (A.B.); dror.sharon1@mail.huji.ac.il (D.S.); 3UMR_S 1112, Laboratoire de Génétique Médicales, University of Strasbourg, CEDEX, 67084 Strasbourg, France; dollfus@unistra.fr; 4Unit of Oculogenetics, Department of Ophthalmology, University of Lausanne, Fondation Asile des Aveugles, 1004 Lausanne, Switzerland; francis.munier@fa2.ch (F.M.); hoaiviet.tran@fa2.ch (H.V.T.); 5Netherlands Cancer Institute, 1066 CX Amsterdam, The Netherlands; m.v.lohuizen@nki.nl; 6Division of Experimental Retinal Therapies, School of Veterinary Medicine, University of Pennsylvania, Philadelphia, PA 19104, USA; wbeltran@upenn.edu

**Keywords:** retinal degeneration, polycomb-repressive complex, neuroprotection, epigenetic

## Abstract

Inherited retinal dystrophies (IRD) are due to various gene mutations. Each mutated gene instigates a specific cell homeostasis disruption, leading to a modification in gene expression and retinal degeneration. We previously demonstrated that the polycomb-repressive complex-1 (PRC1) markedly contributes to the cell death process. To better understand these mechanisms, we herein study the role of PRC2, specifically EZH2, which often initiates the gene inhibition by PRC1. We observed that the epigenetic mark H3K27me3 generated by EZH2 was progressively and strongly expressed in some individual photoreceptors and that the H3K27me3-positive cell number increased before cell death. H3K27me3 accumulation occurs between early (accumulation of cGMP) and late (CDK4 expression) events of retinal degeneration. EZH2 hyperactivity was observed in four recessive and two dominant mouse models of retinal degeneration, as well as two dog models and one IRD patient. Acute pharmacological EZH2 inhibition by intravitreal injection decreased the appearance of H3K27me3 marks and the number of TUNEL-positive cells revealing that EZH2 contributes to the cell death process. Finally, we observed that the absence of the H3K27me3 mark is a biomarker of gene therapy treatment efficacy in XLRPA2 dog model. PRC2 and PRC1 are therefore important actors in the degenerative process of multiple forms of IRD**.**

## 1. Introduction

Inherited retinal dystrophies (IRDs) are caused by at least 300 different mutated genes (RetNet Available online: https://sph.uth.edu/retnet/ (accessed on 27 August 2021)). The initial hit that activates the process of cell death is often specific to each form of disease, such as phototransduction alteration, cilium malformation, or protein misfoldings, etc., but some common mechanisms leading to cell death have now been identified. Indeed, as examples of rod photoreceptor degeneration, cGMP accumulation was observed in early disease stages of several recessive and dominant animal models of retinitis pigmentosa (RP) [1], while cell cycle reactivation with re-expression of CDK4, for example, occurs at a later degenerative stage [2]. Other common pathways were also shown to be activated such as PKG, PARP, Neogenin, microglia activation, Retbindin, etc. [3,4,5,6,7]. So far, it is not clear how these pathways are activated and whether they are connected. We hypothesized that gene regulation changes occurring during the degenerative process could be the common origin of the activation of these pathways. Indeed, several groups have shown by transcriptomic analyses that dozens of genes were differently expressed during the degenerative process [8,9,10]. We have previously demonstrated that the polycomb-repressive complex-1 (PRC1) has a permissive role for the cell cycle activation during retinal degeneration, but we did not investigate the potential role of PRC2, which often interacts with PRC1.

PRC1 and PRC2 are involved during development in controlling cell fate and differentiation as well as morphogenesis by repressing groups of genes [11,12]. EZH1 or EZH2 is a component of PRC2 and trimethylates the lysine-27 of Histone-3 (H3K27me3), initiating gene repression around this mark. PRC1 is then often recruited to the site, through the binding of CBX, followed by the stimulation of RING1B by BMI1. RING1B ubiquitinates H2A (H2AK119Ub), reinforcing gene repression. These epigenetic mark changes can even lead to chromatin compaction (see review [13]).

Among their various roles, polycomb proteins intervene during different stages of the CNS development. BMI1 was shown to control postnatal growth and differentiation of several brain areas, including the cerebellum [14] and the maintenance of the stem cells around the lateral ventricles [2,15]. In the retina, BMI1 loss leads to an alteration of cone development and progressive rod dysfunction revealing the delicate balance of PCR1 in maintaining cell identity or promoting cell death. Concerning PRC2, EZH2 was shown to regulate neurogenesis [16], in part by controlling the fate of Purkinje neurons in the cerebellum [17]. EZH2-deficient mice exhibit a reduction in the number of these neurons and granule precursor cells, and an increased number of GABAergic neurons [17]. EZH2 also controls the duration of the neurogenic period of cortical progenitor cells by maintaining the progenitor identity and neuron differentiation potential [16,17]. In the retina, EZH2 activity and H3K27me3 mark deposition increase during retinogenesis and target several transcription factors [18,19]. Depletion of EZH2 causes microphthalmia, slows retinal progenitor proliferation and accelerates the onset of Müller and rod photoreceptor cell differentiation in conditional KO mice [20,21]. Altogether, these experiments reveal the crucial role of EZH2 to acquire and maintain cell identity during neurogenesis.

A recent study showed a small increase of the H3K27me3 mark in the whole retina of the *Rd1* mice during retinal degeneration [22] but did not document the cell type expressing this mark. In the present study we aimed to focus on the role of H3K27me3 during retinal degeneration, to identify which cell type increases the mark, evaluate the dynamics of this mark change, and temporally position this event in comparison to other biomarkers of retinal degeneration. Different models of retinal degeneration were analyzed to determine whether EZH2 activity is common to different disease forms. We then used EZH2-specific inhibitors to assess EZH2′s contribution to the process of cell death. Finally, we evaluated whether the absence of the H3K27me3 mark can be used as a biomarker of gene therapy efficacy.

## 2. Results

### 2.1. EZH2 Activity Precedes BMI1 Action to Mediate Photoreceptor Death in the Rd1 Retina

We have previously shown that cell death in *Rd1* photoreceptors is tightly dependent on the PRC1-member BMI1 but that its molecular action is independent of the downstream conventional *Ink4a/Arf* pathway [2].

To reveal whether BMI1 loss has an autonomous effect on *Rd1* photoreceptor survival, we induced *Bmi1* deletion specifically and only in photoreceptors, by crossing the conditional *Rd1*;*Bmi1^flox/flox^* mouse with the *Opsin::Cre* line to generate the *Rd1*;*Opsin::Cre; Bmi1^flox/flox^* mice (Figure 1a). In *Rd1*;*Opsin::Cre; Bmi1^flox/flox^* mice, cell type-specific deletion of *Bmi1* was confirmed in photoreceptors by immunohistochemistry for BMI1 at post-natal day 16 (PN16) which corresponds to the stage of advanced degeneration in *Rd1* (Figure 1b). Despite an absence of BMI1 expression in large areas of the outer nuclear layer (ONL) composed of photoreceptors, the decrease of the ONL thickness seen at PN16 in *Rd1* was not rescued in this model (Figure 1b) [1]. This suggests that either BMI1 action is indirect and not intrinsic to photoreceptors or the process of cell death starts before the *Opsin* promoter is active. We thus crossed *Rd1*;*Bmi1^flox/flox^* mice with *Gfap::Cre* mice to delete *Bmi1* in Müller cells. No rescue of photoreceptors was observed in the *Rd1*;*Gfap::Cre*;*Bmi1^flox/flox^* mice either (data not shown), suggesting no or poor contribution of glial cells to photoreceptor survival in the *Rd1;Bmi1^−/−^* animals.

As PRC2 often acts upstream of PRC1 to regulate gene expression, we hypothesized that PRC2 may already be involved in this death process, so we investigated H3K27me3 binding, the PRC2-specific histone posttranslational modification (HPTM). EZH2 belongs to PRC2 and mediates the trimethylation on Lysine 27 of Histone3 (H3K72me3) to favor gene repression in association with PRC1 through BMI1 and RING1b [2,3]. Careful analysis of retinal sections during degeneration of PN16 *Rd1*;*Opsin::Cre;Bmi1^flox/flox^* mice revealed an accumulation of H3K27me3 marks in some cells in the ONL (Figure 1c). These data show that despite *Bmi1* function loss in postmitotic photoreceptors, PRC2 is mobilized during the degenerative process, and that cells in the ONL undergo marked epigenetic modifications. We thus followed the H3K27me3 mark pattern during *Rd1* retina degeneration.

### 2.2. Enhanced H3K27 Trimethylation (H3K27me3) in Rd1 Photoreceptors and Other IRD Models

In PN12 *Rd1* mice, immunohistochemistry against H3K27me3 showed positive cells in the retinal ganglion and inner nuclear layers and a robust nuclear labeling in several nuclei composing the outer nuclear layer (ONL) (Figure 1d). The staining was heterogeneous with various distributions in the nuclei (Figure 1d,e) and the number of H3K27me3-positive cells in the ONL varied temporally with a peak at PN13 (Figure 1f). We noted also that some ONL areas have a low level of H3K27me3 in comparison to the WT level. The presence of the high level of the H3K27me3 (H3K27me3^h^) mark at PN12 is more abundant in the central retina in comparison to the periphery, which is just achieving retinogenesis (Figure 1g). These observations suggest a highly dynamic state of this mark. H3K27me3^h^ co-localized in cells expressing the rod photoreceptor-specific protein, RHODOPSIN (Figure 1h), attesting that this mark is present in these sensory cells. In order to isolate photoreceptor cells from other retinal cells, we used the Crx-GFP mouse strain which expresses GFP specifically in both rod and cone photoreceptors (Figure 1i,j). Protein analyses of photoreceptor cells isolated from *Crx-GFP* (WT) and *Rd1-Crx-GFP* retina at PN12 revealed that EZH2 was increased in degenerating sensory cells (Figure 1k). Surprisingly, the total level of H3K27me3 in the retina at PN12 is lower in the *Rd1* mouse compared to WT, highlighting the general low level of H3K27me3 observed in the retina by immunohistochemistry, although some cells have a high expression level of the mark. In contrast, no H3K27me3 dense mark was observed in the ONL of the WT retina (Figure 1d) as previously described for mature WT retina [4].

To determine if the variation of distribution of the H3K27me3^h^ mark in *Rd1* retina is specific to this mark or reveals general deregulations of HPTMs homeostasis, we examined the pattern and distribution of H3K9me2 and H3K4me3 marks. After staining, unlike for H3K27me3^h^, we did not detect any obvious differences for H3K9me2 nor H3K4me3 marks between the WT and *Rd1* ONLs at PN12 (Figure 2a–c). Altogether, these results strongly suggest that the *Rd1* rodent’s photoreceptors undergo specific epigenetic remodeling with H3K27me3 accumulation during the degenerative process.

The *Rd10* mouse is another retinal degeneration model of *PDE6β* deficiency, as the *Rd1*, but the mutation is different [5,6]. In the *Rd10*, while the degeneration remains fast (in our colony), the onset of retinal cell loss is delayed and occurs when the retina is mature (PN18). The number of TUNEL-positive cells in the central retina peaks at PN22–23 and differs from *Rd1* in which half of the photoreceptor cell death occurs in the ONL by PN13–14 [7]. Interestingly, staining of *Rd10* retina sections for H3K27me3 also showed a distinct increase of the mark at PN12 (Figure 2d, bottom panel). A similar observation was made in a large animal model of PDE6B deficiency, the *Rcd1* dog (Appendix A). In addition, we observed a strong presence of the H3K27me3^h^ mark in the ONL of two models of autosomal dominant RP, the *Rho^P23H^* mouse and the *Rho^S334ter^* rat (Figure 2d, top and middle panels respectively), as well as in the *Rhodopsin*^−/−^ mice with slow degeneration kinetics (data not shown). Two transgenic mouse lines mimicking retinal degeneration associated with ciliopathy of the photoreceptors, such as the *Fam161a^tm1b/tm1b^* (*Fam161a* knockout) mouse and the Bardet Biedl Syndrom-10 (BBS10 KO mice), also presented isolated photoreceptors with high H3K27me3 level (Figure 2d). The ONL of a dog model of ciliopathy, the X-linked retinitis pigmentosa, also contained cells positive for H3K27me3^h^ (see Section 2.6). Thus, our results in nine animal models suggest a common mechanism causing H3K27me3 that is shared in a variety of non-allelic IRDs Table 1).

### 2.3. H3K27me3 Accumulates in a Human Retinitis Pigmentosa (RP) Patient Retina

To better implement the relevance of the H3K27 hypermethylation in retinal disorders, we investigated whether this phenotype could also occur in human retina. The retinas from one healthy donor and one human RP patient (85-year-old) affected by a dominant mutation in the *Rhodopsin* gene (Gln29Arg) were labeled for H3K27me3^h^. The eye enucleation was performed because of a Melan-A-positive melanoma in the choroid. The eye characterization was performed outside the tumor area. The number of cells containing the H3K27me3^h^ mark was noticeably enriched at the periphery of the RP retina, in the areas where the degenerative process started initially, and also in the near peripheral area (Figure 2f) compared to the human healthy donor sample (Figure 2e). The RP patient was diagnosed before the enucleation. By contrast, only few cells strongly positive for the H3K27me3 mark were observed in the central RP retina, which is coherent with the process of RP retinal degeneration starting at the retina periphery. We also noted that the H3K27me3 mark was present in the healthy photoreceptors but was homogenously expressed with no accumulation of the mark. Together, these data spot the H3K27me3^h^ as a hallmark of the degenerative process in RP and may be linked to retina areas with ongoing photoreceptor cell death.

### 2.4. H3K27me3 Accumulation in the Rd1 Retina Precedes Late Events of Photoreceptor Death

To gain better insight into the temporal manifestation of H3K27me3^h^ in *Rd1* retina, we evaluated the time course of this mark, starting from PN5, prior to disease onset, until PN15, when only less than 30% of photoreceptor cells remain in the ONL. H3K27me3^h^ was already detectable at PN8 (Figure 1f) and its temporal expression pattern was tightly linked to the degenerative course already described [1,8]. The number of cells with H3K27me3^h^ increased with time until PN13, which coincides with marked photoreceptor loss in *Rd1* retina. Moreover, we noticed that the H3K27me3^h^ signal was primarily enriched in the central part of the retina in the region flanking the optic nerve (PN8–12), and then onwards toward the periphery, as the disease progresses in this mouse model (Figure 1g, data shown only at PN12). Interestingly, Western blot analysis of purified *Rd1*; *Crx*-GFP-positive photoreceptors revealed an increase of the PRC2 component EZH2 protein at PN12 (Figure 1k). Together, these data suggest that H3K27me3 accumulation strongly correlates with the disease progression in the *Rd1* mice and may be the consequence of EZH2 upregulation.

The course of cell death in photoreceptors is orchestrated by a compilation of distinct molecular events [9]. In *Rd1* rod photoreceptors, cGMP accumulation commonly occurs during the very early stages of photoreceptor cell death cascade, while Terminal dUTP strands break labeling (TUNEL) reveals the end point when cells undergo their DNA fragmentation phase. As the first appearance of high H3K27me3 coincides with the time frame of cGMP intracellular increase in the *Pde6β* null mouse retina [10], we asked whether cGMP and H3K27me3^h^ co-exist in the same cell or not. We did not observe any cells in which the cGMP staining co-localized with H3K27me3^h^ in *Rd1* retinas at PN12 (Figure 3a top panel), suggesting that H3K27me3 is a downstream event in regard to the cGMP accumulation described as a very early marker of photoreceptor homeostasis deregulation (29). Among the H3K27me3^h^ photoreceptors, 60% were also TUNEL-positive, whereas 40% of the TUNEL-positive population co-localized with H3K27me3^h^, suggesting a relationship between H3K27me3^h^ and cell death in this mouse model (Figure 3a bottom panel, Figure 3b), but revealing two distinct cell stages, suggesting that H3K27me3 precedes DNA fragmentation. In addition, 65% of the H3K27me3^h^-positive cells were also positive for CDK4 at PN12 (Figure 3a middle panel, Figure 3c), which has been shown to be a late marker of photoreceptor degeneration [1]. Approximately 85% of CDK4-positive cells were also H3K27me3^h^-positive and the time course analysis of H3K27me3^h^ and CDK4 demonstrated an elevated number of cells positive for H3K27me3^h^ before the appearance of CDK4-positive cells (Figure 3d). Altogether, these results reveal successive emergences of H3K27me3^h^ and CDK4 prior cell death (TUNEL positive) suggesting that EZH2 is involved in an intermediate stage of retinal degeneration (Figure 3e).

### 2.5. EZH2 Contributes to the Process of Cell Death

Two different patterns of H3K27me3^h^ were observed during retinal degeneration, one corresponding to an accumulation of this mark in a subset of photoreceptors, whereas others showed reduced expression. In this study, we challenged the H3K27me3 accumulation event with an EZH2 inhibitor to reveal whether this enzyme contributes to the process of cell death. We used the UNC1999 EZH2 inhibitor to target its catalytic SET domain [11]. The effective doses were first tested in a proliferating prostate cancer cell line (PC3) in which EZH2 was shown to be active [12]. At concentrations of 5 and 10 µM, UNC1999 was capable of blocking the endogenous levels of H3K27me3 compared to DMSO control (Figure 4a,b), thereby suggesting the efficient doses for in vivo investigation. Based on this result, the *Rd1* (PN8) retinas were then exposed to UNC1999 by intravitreal injection to an estimated concentration of 5 µM. The efficacy of treatment was evaluated by monitoring the H3K27me3^h^ mark and the extent of cell death using TUNEL staining on sections from mice sacrificed 4 days post-injection. The hyper H3K27me3 signal was preserved in DMSO-treated eyes, whereas such signal was strongly reduced (by 75%) in the ONL of UNC1999-injected retinas (Figure 4c,e). Interestingly, the number of TUNEL-positive cells was also substantially lowered, by about 70%, in the corresponding area compared to DMSO-treated controls (Figure 4d,f). Similar observations were obtained with the *Rd1* mice injected with another EZH2 inhibitor, EPZ6438 (Appendix A).

The *Fam161a^tm1b/tm1b^* knockout mouse is another disease model of RP (RP28) with a relatively slow retina degeneration resulting in the loss of around 80% of photoreceptors by 6 months of age [13]. Similar to the *Rd1* mice, we administered a dose of UNC1999 to 2-month-old *Fam161a^tm1b/tm1b^* mice but repeated the treatment every week for one month until the animals were sacrificed 4 days after the 4th intravitreal injection. The retinal sections were analyzed for photoreceptor cell survival and H3K27me3 presence along the dorso-ventral axis (Figure 4g,h). Our data show a significant preservation of the number of rows in the ventral ONL, where the needle was located during the injection. No significant difference was observed in the dorsal compartment. We did not observe any significant decrease of the H3K27me3^h^ levels between the injected and non-injected side, with the exception of a local increase in the dorsal part of the UNC1999-treated group. A general tendency of the mark accumulation was observed probably because of the reactivation of the EZH2 enzyme after the drug elimination. Collectively, these data strongly suggest that EZH2 contributes to the process of cell death and that inhibiting EZH2 activity in vivo delays photoreceptor degeneration in mouse models of retinal degeneration.

### 2.6. The H3K27me3 Mark Is Absent in RPGR-Mutant Dogs after Gene Augmentation Therapy Treatment

Ocular gene therapies to treat different forms of IRDs have shown great success to restore visual functions and associated behaviors (for review, see [14]). However, it is not always clear whether the process of retinal degeneration is fully stopped and if a true cure was developed [15,16,17]. To approach this question, we investigated the expression of the H3K27me3 mark after gene therapy delivered at early (5 weeks of age), mid- (12 weeks of age), and late-stage of the disease (26 weeks of age) in the XLPRA2 dog, a canine model of X-linked retinitis pigmentosa (XLRP) caused by a two nucleotide deletion in exon ORF15 of the Retinitis Pigmentosa GTPase Regulator (*RPGR*) [18,19]. We have previously shown that an AAV2/5-RPGR vector carrying a stabilized form of the human *RPGR* cDNA (*hRPGRstb*) [20] can positively modify the disease course in this model by preserving the retina integrity and function [21]. We used archival retinal sections from this earlier study to analyze to which extent the process of retinal degeneration was changed by this gene therapy. In non-treated *RPGR* mutant dogs, H3K27me3^h^ mark was detected at 5 and 12 weeks of age and reduced at 26 weeks of age, yet levels were higher than that observed in a 26-week-old WT dog (Figure 5e).

To evaluate EZH2 activity in AAV2/5-*RPGRstb*-injected animals, we used retinal sections that encompassed treated and non-treated areas. The treated areas were identified on the basis of transgene (*hRPGRstb*) expression detected by IHC, a thicker ONL, and the outer segment preservation (Figure 5a). We compared the treated areas to the non-treated ones, which were of similar sizes (for details, see the method section). Animals treated at 5, 12, and 26 weeks of age had H3K27me3^h^ positive cells in the ONL at the time of termination (113 weeks of age) in all non-treated areas, whereas no H3K27me3^h^-positive photoreceptor cells were present in the AAV-treated areas of these same retinas (Figure 5f). These observations strongly suggest that delivery of an adequate amount of a stable therapeutic vector provided long-term (~2 years) interruption of the degenerative process.

## 3. Discussion

BMI1 was shown to play a major role during retinal degeneration [1] and to better characterize its action, we deleted its expression specifically, either in the photoreceptors or the glial cells using a CRE recombinase approach. However, in contrast to the rescue observed in the *Rd1:Bmi1* knockout model, none of the transgenic lines tested for photoreceptor or glial cell targeting showed photoreceptor protection. One explanation may be related to the expression timing of the CRE activation driven by the *Opsin* promoter, which is relatively late during the photoreceptor maturation. Indeed, *Pde6b* is expressed very early during photoreceptor development, well before the *Opsin* gene [22]. Thus, the absence or low levels of PDE6b in the early developmental stage of photoreceptors may already initiate alterations in these cells. In line with this hypothesis, we observed an early accumulation of the H3K27me3 mark in the degenerating photoreceptors (from PN10) indicating that PRC2 is mobilized during the process of cell death. This suggests that EZH2 probably precedes and/or co-opts with BMI1 action, which may also contribute to the cell death process before the expression of the opsin.

Interestingly, the time course of the H3K27me3 epigenetic mark accumulation in the photoreceptors seems to poorly correlate with the appearance of cGMP accumulation. The epigenetic mark increased before the surge of CDK4 and TUNEL-positive cells, suggesting that EZH2 is activated after first events of cell stress and may participate to the process of cell death. Nonetheless, around 60% of the H3K27me3^h^-positive cells are also TUNEL-positive, suggesting a quite long-lasting action of EZH2, which is in line with the slow process of photoreceptor degeneration of around 80 h described previously [8].

In a short-term experiment, we demonstrated that EZH2 is indeed involved in the cell death activation. Intravitreal treatments of *Rd1* eyes with an EZH2 inhibitor, before the peak of cell death, markedly and significantly decreased the appearance of the H3K27me3^h^ mark and reduced the number of cells entering in a cell death process. A previous study also revealed by Western blotting an increase of the H3K27me3 mark in the retina during retinal degeneration [23], but without determining which cells are involved and whether the increase is progressively uniform or not. We observed that only scattered photoreceptors at a defined time showed a marked H3K27me3 accumulation. Our experiments suggest that the *Rd1* photoreceptors are in a steady state compatible with cell survival until they reach a stress level that activates EZH2 leading to an abnormal increase of the H3K27me3 mark. If this hypothesis is true, cell death could be markedly postponed if a treatment is provided in a repeated manner, without disturbing the endogenous activity of EZH2. The results obtained with the *Fam161a* knockout mice tend to support this observation when performing multiple injections. However, so far, the effect is modest probably due to the drug’s short half-life, a rapid vitreal clearance, and the lack of homogeneous distribution throughout the retina. Indeed, the number of H3K27me3-positive cells was not diminished 4 days after the last injection, indicating a rapid loss of the injected inhibitor. In Sahaboglu et al. [24], the authors estimated the half-life of olaparib to be eight minutes (which is a larger molecule than UNC1999), strongly supporting a rapid clearance of the EZH2 inhibitor. Our results also revealed that EZH2 hyperactivity rapidly recovered when the inhibitor is absent. An ocular formulation has to be developed to improve the drug efficiency and distribution.

In the Zheng et al. study [23], DZNEP was shown to protect the *Rd1* retina when administrated at PN0, but this treatment affects the differentiation of the rods, committing them to a cone fate, as suggested by the authors. To support this hypothesis, differential gene expression analyses of the DZNEP-rescued retina highlight the PI3K pathway, which was also identified in a recent study of cone neuroprotection [25]. Moreover, DZNEP was shown first as an inhibitor of the S-adenosylhomocysteine hydrolase, to be efficient on EZH2 only at high doses and is not recognized as a specific molecular probe [26]. The use of EZH2 inhibitors such as UNC1999 and EPZ6438 in our study clearly shows that EZH2 contributes to the photoreceptor death activation.

EZH2 is involved at the end of retina development by maintaining neurogenesis. The conditional deletion of EZH2 in retina progenitor cells (Pax6-Cre or DKK3-Cre) accelerates differentiation and decreases retinogenesis duration leading to a deficit in late-appearing neurons such as rods and bipolar cells [27,28]. EZH2 is also involved in the cell survival maintenance once retinal cells are formed. The conditional knockout of the *Ezh2* gene at E11.5 leads to a progressive degeneration of the photoreceptors related to a reactivation of the Six1 gene which is expressed in retinal progenitors [29]. The degeneration is slow, reducing by around 50% the photoreceptor layer over one year. Thus, EZH2 appears to be a key regulator of retinal cell fate acquisition and maintenance. Our study showed that the pathological EZH2 hyperactivity observed during retinal degeneration appears to markedly alter photoreceptor survival.

The present study focused on photoreceptor degeneration, which degenerative processes are different to those documented for the retinal ganglion cells [30,31]. It would be nonetheless interesting to reveal whether EZH2 pathway or other epigenetic modifications occurs during glaucoma or other models of optic neuropathies [32,33]. The use of the *Oct4* (also known as *Pou5f1*), *Sox2* and *Klf4* genes (OSK) to reverse DNA methylation during glaucoma and restore vision [34] strongly suggest that PRC2 may be involved in glaucoma as well. EZH2 hyperactivation was also observed in other models of neurodegeneration such as the ataxia-telangiectasia mouse [35] and in susceptible neurons of mouse models of Alzheimer’s disease [36]. In the ataxia-telangiectasia mouse model, the delivery of a lentiviral vector coding for a shRNA against EZH2 into the cerebellum protects the Purkinje neurons from degeneration [35]. Interestingly, the action of EZH2 was reported to be linked to epigenetic modifications of H3K27me3 occupancy on genes involved in cell cycle, cell death, and neuron development. These pathways are also activated in the degenerating photoreceptors, but the link with EZH2 remains to be determined.

Since the H3K27me3 mark accumulation revealed the degenerative process of photoreceptors in various models of inherited retinal dystrophies, we investigated whether this mark could serve as a common biomarker for therapies aiming at rescuing and/or restoring retinal functions. Using material available from previous experiments, we thus analyzed the retina of dogs bearing mutations in the ORF15 of the *RPGR* gene and treated by gene augmentation therapy at different times of the disease progression to restore the gene function [19]. The study had demonstrated that early or delayed treatments with AAV2/5-*hRPGRstb* restored retinal function and preserved retina integrity. The present results clearly show that in all treated areas of all dogs, no accumulation of the H3K27me3 mark was detected, strongly suggesting that no processes of cell death were progressing in RPGR-expressing photoreceptors. This robustly supports the treatment’s long-term efficacy and reveals that the absence of the H3K27me3^h^ mark can serve as a biomarker of efficient therapies.

Epigenetic studies of the marked genome landscape integrated with gene expression analysis will indicate whether the appearance of the epigenetic mark correlates with a deep change of gene expression. Such analyses should disclose when a critical disease stage is reached, rendering more difficult a treatment in a chronic situation. Nonetheless, the stochastic appearance of isolated photoreceptors with the accumulation of the H3K27me3^h^ mark suggests that the disease progresses heterogeneously within the tissue and that homeostasis deregulation, at a certain time point, drives the photoreceptor cell to switch its genetic program to a cell death fate. This also suggests that modulation of this event can be undertaken at any time during the degenerative process in order to preserve the remaining photoreceptors. The fact that nine animal models of retinal degeneration show a similar pattern of H3K27me3 accumulation is encouraging in that perspective. Long-term studies of EZH2 inhibitor application are needed to refute or confirm the therapeutic potential of this gene agnostic approach.

## 4. Materials and Methods

Mice and livestock: The FVB-WT, FVB-*Rd1* (both from Charles-River)*,* FVB-*Rd1;Bmi1^+/−^*, *Rd1*;*Bmi1^flox/flox^* (both from the Netherland Cancer Institute, Amsterdam, Netherland), *Opsin::Cre, Gfap::Cre*, Crx-GFP (Jackson Laboratory, Bar Harbor, ME, USA) as well as the *Fam161a^tm1b/tm1b^* knockout mouse (from the Hadassah medical center; Jerusalem, Israel) mice were maintained in an animal facility with a 12h:12h light/dark cycle. All the experiments as well as the procedures were approved by cantonal veterinary authorities. The animals were treated according to institutional and national rules as well as Association for Research in Vision and Ophthalmology (ARVO) guidelines.

Chemicals: Two EZH2 inhibitors were tested in this study, UNC1999 (TOCRIS, Minneapolis, MN) and the EPZ6438 (MedChem, Monmouth Junction, NJ, USA). UNC1999 was dissolved in 10% DMSO diluted in PBS. The stock solution at 10 mg/μL was kept at −80 °C. EPZ6438 was already conditioned in a saline solution at 10 mM containing 10% of DMSO.

Anesthesia: Mice were anesthetized with a mixture of Ketalar (0.6 mg/10 g) and Domitor (10 μg/10 g) in 200 μL of NaCl 0.9%. After the operation, Antisedan (10 μg/10 g) was given not before 30 min after the anesthesia as reversal solution.

Intravitreal injection: Once the mouse is anesthetized, the eye was exorbitated and maintained by a thread and a knot which surrounded the globe. A first incision was practiced at the level of the *ora serrata* with the needle of an insulin syringe. The knot was then slightly relaxed to reduce pressure on the eye. A blunted needle (34G) mounted on a Hamilton syringe was introduced through the incision into the vitreal cavity and the solution (1 μL) was slowly injected close to the retinal surface.

Cell culture: The PC3 human prostate cancer cell line was a generous gift from Clem Babarbouchky (The Rockfeller University, New-York, NJ). 200,000 cells were seeded in 12-well plates in the DMEM medium with 10% of FBS. After 3 days in culture, the EZH2 inhibitor was added at concentrations of 2.5 to 10 μM and the cells processed for Western Blot 3 days later.

Histology and tissue processing.

The retinas were fixed with 4% paraformaldehyde (PFA) in PBS for 60 min at room temperature (RT), bathed in successive PBS-sucrose concentrations (10% for 2 h, 20% for 2 h and 30% sucrose overnight at 4 °C). The tissue was then embedded in Yazulla and frozen at −20 °C before sectioning.

Isolation of photoreceptors from WT and *Rd1* mice.

Crx-GFP and FVB-*Rd1* mice were crossed to obtain WT-Crx-GFP and Rd1-Crx-GFP mice. Groups of animals were sacrificed at PN12 and the retina was then isolated immediately and incubated for 30 min in reduced serum medium, OpTi-MEM (Life Technologies, Carlsbad, CA, USA). Four to six retinas of mice from the same litter were pooled and subjected to papain I dissociation using the Papain dissociation kit (Worthington biochemical, Lakewood, NJ, USA) according to the manufacturer’s instructions. The sample preparation was further loaded onto the column of MoFlo Astrio system (Beckman Coulter’s company at the UNIL platform, CHUV, Switzerland) equipped with a 488 nm green laser to excite GFP. GFP-positive photoreceptors were sorted using constant excitation and collected in PBS, snap-frozen, and immediately stored at −80 °C until the day of analysis.

Terminal dUTP strand break labeling: Cell death was monitored with the in situ cell death detection Kit (Cat: 12156792910. Roche diagnostics; Rotkreuz, Switzerland) and the terminal dUTP Nick-ends labeling (TUNEL) was performed according to the manufacturer’s instructions.

Epitope retrieval before H3K27me3 staining: frozen slides were warmed at room temperature, and then washed 3 times with PBS. The slides were placed in a plastic container and filled up with the retrieval buffer (10 mM citric acid buffer, pH 6.0) and transferred to a microwave oven. The samples were heated at 600 W for approximately 2 min until the liquid started boiling. The buffer was permanently refilled and boiled again sequentially for a total period of 5 min. Alternatively, the samples were heated at 96 °C for 13 min in an oven. The sections were then cooled slowly in citric buffer at room temperature for 30 min and rinsed thoroughly with PBS. Slides were then processed for immunohistochemistry as described below.

Immunohistochemistry: Immunohistochemistry was performed on 14 µm cryogenic sections. The frozen sections were thawed at room temperature and re-hydrated with PBS before being permeabilized for 15 min with Triton X-100. Sections were blocked with 3% bovine serum albumin, 5% normal goat serum in PBS containing 0.3% Tween 20. Slides were incubated overnight at 4 °C with the solutions prepared in blocking buffer (see Table 2 for antibody concentration and origin). Then, the antibody solution was discarded, and the slides were washed 3 times with PBS. The secondary antibodies applied for 1h at RT were prepared in PBS as follows: goat anti mouse or goat anti rabbit 1:1000 and 1:2000 for Alexa Fluor^®^ 488 and Alexa Fluor^®^ 633, respectively. Following three washes, the slides were incubated with DAPI solution (1:1000 in PBS to stain the nuclei). Finally, the slide was mounted with a Mowiol (Merk-Millipore, Burlington, MA, USA) solution and the signal was visualized with fluorescence microscopy.

Cell counting in the RPGR-XLRP mutant dog retina slices: The number of highly H3K27me3-positive cells in the ONL was counted in the AAV2/5-hstbRPGR treated area, versus the non-treated area. For dog Z412, the equivalent of six 20× fields of treated area versus six 20× fields of non-treated area were analyzed; for dog Z459, three 20× fields in the treated area versus seven untreated and for the Z460, 7 20× fields were investigated for each treated and non-treated area.

Preparation of tissue homogenate: Freshly dissected retinas from WT and *Rd1* mice were homogenized in lysis buffer made of 50 mM Tris-HCl pH 7.6, 150 mM NaCl, 1mM EDTA, 0.25% Triton X-100, and 0.25% Nonidet NP40, and protease inhibitors cocktail (Sigma, Saint-Louis, MO, USA). Subsequent sonication step at 4 °C was performed to release the components of the nuclei. Cytoskeleton and debris were pelleted at 14,000× *g* for 10 min and the cleared lysate collected, transferred into small aliquots in fresh Eppendorf tubes and stored at −80 °C until the day of analysis.

Western blotting: Equal amounts (20 μg) of tissue homogenate were resolved in 12% polyacrylamide gel electrophoresis under constant voltage. The proteins were then transferred onto a polyvinylidene (PVDF) membrane using a wet transfer system (BioRad, Cressier, Switzerland). The PVDF membrane was blocked for 1 h with 5% non-fat dried milk in PBS at room temperature and then incubated overnight with the primary antibody diluted in 5% non-fat dried milk, PBS 0.1% Tween 20 (PBST) as indicated (see Table 1). The membrane was washed 3 times with PBST and further incubated 1 h at room temperature with horseradish peroxidase (HRP)-conjugated secondary antibody diluted in PBST. The last washes were carried out twice with PBST and once with PBS. Finally, the high-sensitivity chemiluminescent HRP substrate (Witec ag) was added and the bands revealed with Fujifilm chemiluminescent cassette in dark room.

Statistics: All the data are expressed as mean plus or minus the standard error of the mean (SEM). When two groups were compared, data were analyzed using the Mann–Whitney U-test and the *p*-value given.

## 5. Patents

Patent application PCT/EP2019/067784 in national phase entry: “Inhibition of PRC2 subunits to treat eye Disorders” Y. Arsenijevic & M. Mbefo.

## Figures and Tables

**Figure 1 ijms-22-09331-f001:**
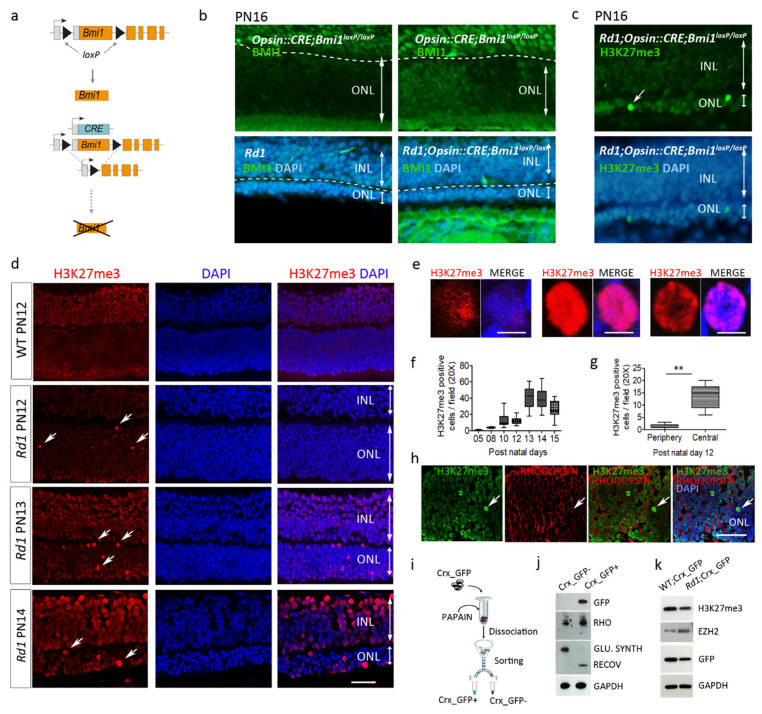
Photoreceptor degeneration is associated with increased H3K27me3 mark in Rd1 mice. (**a**,**b**) Conditional Bmi1 deletion driven by the photoreceptor-specific opsin promoter has no effect on photoreceptor degeneration in *Rd1*. (**a**) Schematic representation of genetic manipulations performed on *Rd1;Opsin::CRE**;Bmi1^loxP/loxP^* mice. Black triangles indicate *loxP* sites that surround the *Bmi1* gene. (**b**) The *Opsin::CRE**;Bmi1^loxP/loxP^* animal shows a specific deletion of *Bmi1* within the ONL in a mosaic pattern (2 top panels). No rescue of photoreceptors was seen in *Rd1;Opsin::CRE**;Bmi1^loxP/loxP^* at PN16 compared to the *Rd1* control mice (2 bottom panels). (**c**) Accumulation of the H3k27me3 mark was observed in some degenerating *Rd1;Opsin::CRE**;Bmi1^loxP/loxP^* photoreceptors at PN16. (**d**) The abundance, distribution (white arrows) and nuclear localization of high H3K27me3 (H3K27me3^h^)-positive cells are age-dependent in *Rd1* retina (red staining). (**e**) Diverse patterns of H3K27me3^h^ expression in the photoreceptor nuclei during the degenerative course. (**f**) The number of H3K27me3^h^ positive cells in *Rd1* mice increases from PN8 with the severity of the disease, peaks at PN13 and was shown to initiate in the central retina (**g**). (**h**) H3K27me3 accumulation occurs in rod cells, as demonstrated by H3K27me3^h^ and RHODOPSIN (Rho, red labelling, arrows) staining colocalization. The dissociation and sorting (**i**) of the GFP-positive cells allows to isolate photoreceptors from the *Crx-GFP* models, as confirmed by Western blot analysis (**j**). (**k**) Western blot of isolated photoreceptors from the *Crx-GFP* and *Rd1;Crx-GFP*
*PN12* retina reveals increased expression of EZH2 in degenerating photoreceptors. Histograms and plots are from an average of at least 8 sections, n = 4 animals per group. Statistical analysis: ANOVA t test, *p* < 0.01 (**). Error bars indicate SEM. ONL = outer nuclear layer, INL = inner nuclear layer. Scale bars = 40 μm (**d**,**h**).

**Figure 2 ijms-22-09331-f002:**
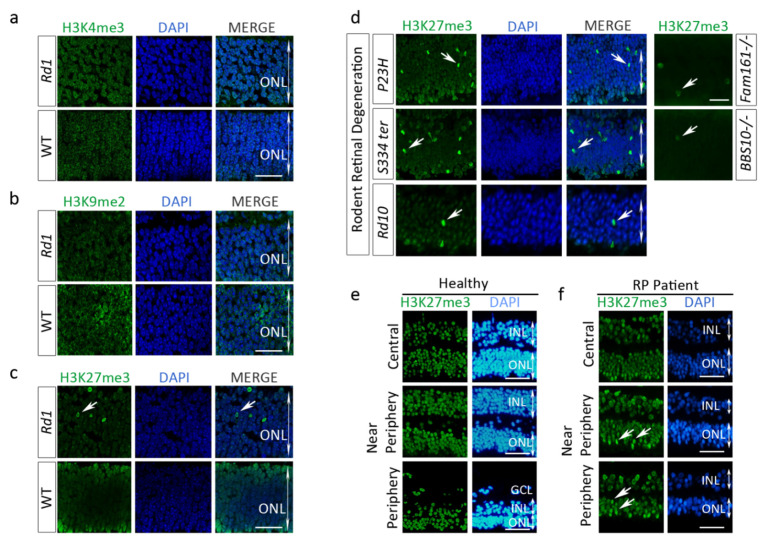
High epigenetic mark accumulation is specific to H3K27me3 in murine models of retinal degeneration and high level of H3K27me3 mark is also detected in human photoreceptors from one retinitis pigmentosa (RP) patient. (**a**,**b**) No obvious differences in H3K4me3 or H3K9me2 marks were observed between WT and *Rd1* retinas at PN12, whereas some photoreceptor nuclei expressed H3K27me3^h^ in the *Rd1* retina (**c**). (**d**) H3K27me3 accumulation was also observed in two rodent models of a dominant form of inherited retinal degeneration (mouse rhodopsin *P23H* and rat *S334ter*) as well as two mouse models of photoreceptor ciliopathy (*Fam161a^−/−^* and *BBS10^−/−^*), and in another model of PDE6B deficiency, the *Rd10* mouse. Retina sections from one human healthy donor (**e**) and one RP patient (**f**) with moderate degeneration in the central retina and advanced cell loss in the periphery. Increased H3K27me3^h^ was much more predominant in areas with advanced retinal degeneration (periphery, arrows). Note that in retina areas with normal photoreceptor architecture, few nuclei also contain the H3K27me3 mark (arrows). Calibration bar: 40 µm (**a**–**c**) and 40 µm for (**e**,**f**), respectively.

**Figure 3 ijms-22-09331-f003:**
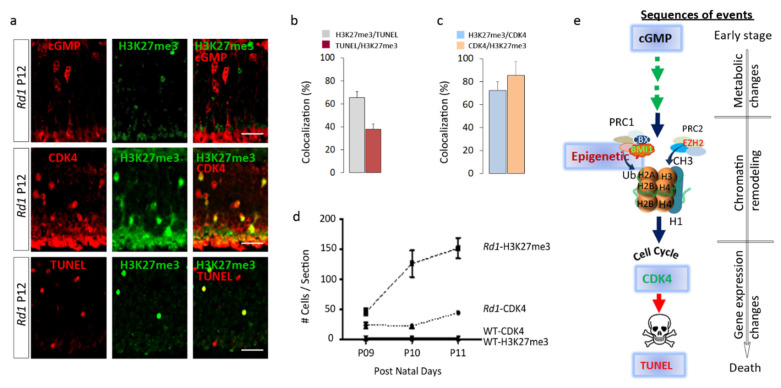
H3K27me3^h^ mark increase is an intermediate event between the cGMP accumulation and DNA fragmentation (TUNEL) in *Rd1* mice. (**a**) Photoreceptor cells in *Rd1* retina were double stained with H3K27me3 and either cGMP (top panel), CDK4 (middle panel) or TUNEL (bottom panel). No co-localization was seen with cGMP and H3K27me3, whereas fractions of CDK4 and TUNEL-positive cells also contained H3K27me3. (**b**) Bar graph representing the percentage of cells with H3K27me3 and TUNEL or (**c**) CDK4 at PN12. Note that less than 40% of the TUNEL-positive cells are also H3K27me3-positive (*n* = 4). (**d**) Time course analysis of CDK4 and H3K27me3 in *Rd1* and WT mice at PN 09, 10, and 11 (*n* = 4 animals per age). The number of H3K27me3-positive cells peaks before CDK4 expression. (**e**) We propose this sequence of events during retinal degeneration (no evidence now of functional relationship). Scale bar = 20 µm. Error bars indicate SEM.

**Figure 4 ijms-22-09331-f004:**
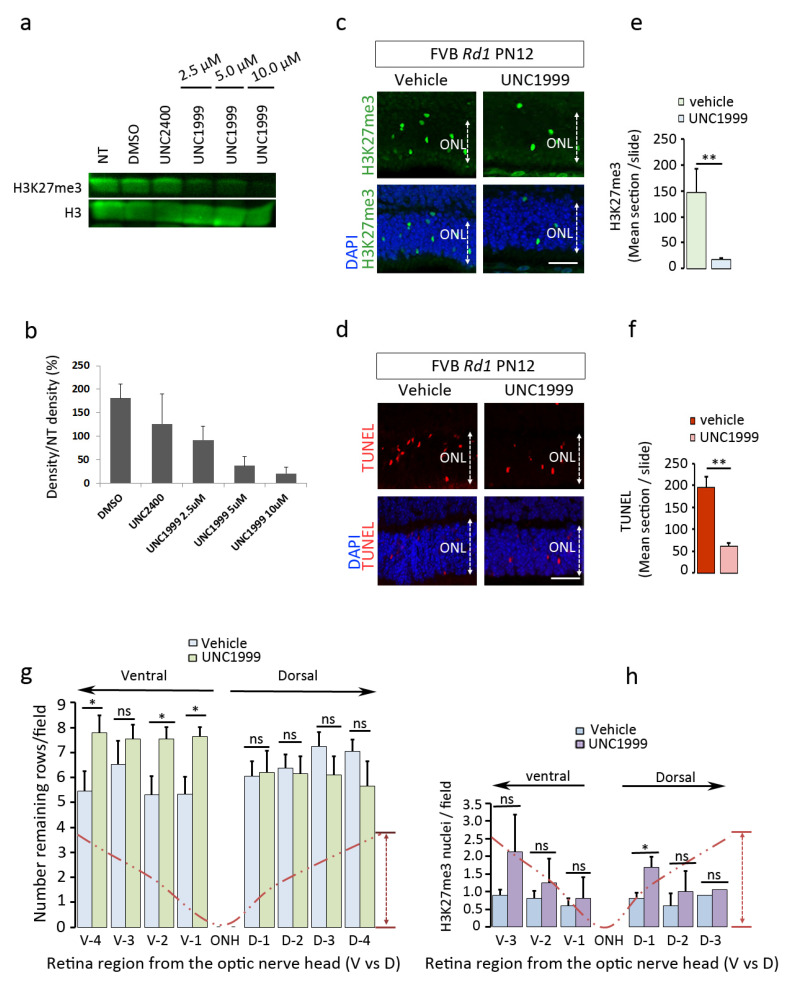
Pharmacological inhibition of EZH2 in vivo delays photoreceptor cell death in *Rd1* and *Fam161a^tm1b/tm1b^* mice. (**a**) PC3 cells were treated with increasing doses of the EZH2 inhibitor UNC1999 as indicated and Western blot quantification (**b**) revealed that 5 and 10 µM were the most effective doses leading to a distinct reduction of the H3K27me3^h^ mark. The UNC2400 inhibitor, which has 1000 times less potent activity to inhibit EZH2, showed no significant effects at 10 µM. (**c**–**f**) Rd1 eyes were injected at PN8 with the vehicle (DMSO) or UNC1999 (2.5 µM final eye concentration). UNC1999 induced a reduction of H3K27me3^h^ mark (**c**,**e**) and TUNEL (**d**,**f**) positive cells. (**g**,**h**) Similar experiment was performed in Fam161a^tm1b/tm1b^ mice at 2 months of age with 4 injections at an interval of one week each. The animals were analyzed 4 days after the last dose. Data indicate a statistically significant preservation of photoreceptor rows along the ventral axis of the degenerating retina (**g**), although no global changes were observed for the H3K27me3^h^ mark in the same area, with the exception of the D1 area showing a significant increase of the mark (**h**). Statistical analysis: two-way ANOVA, *p* < 0.05 (*), n = 5 retinas analyzed per condition for (**g**) and Mann–Whitney analysis was performed for (**e**,**f**), n = 3 to 4, *p* < 0.01 (**), ns: not significatif. ONL = Outer Nuclear Layer.

**Figure 5 ijms-22-09331-f005:**
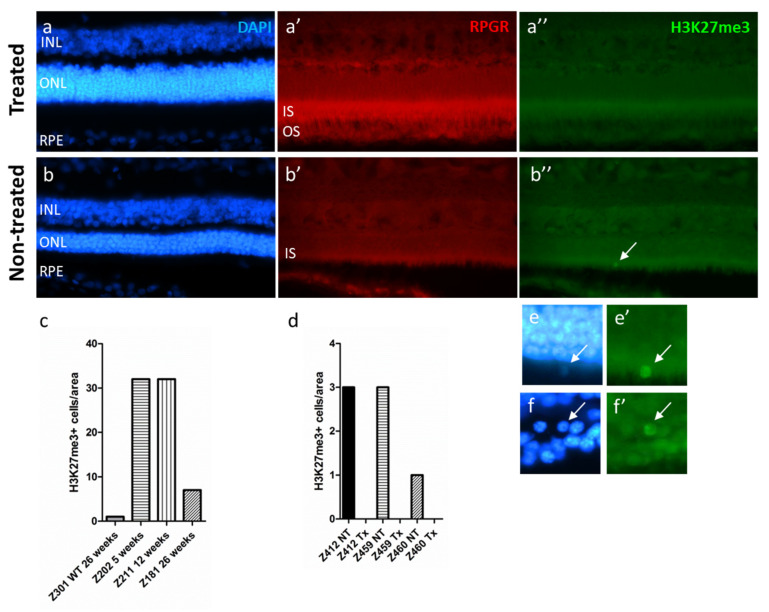
Absence of H3K27me3^h^ mark following gene therapy in *RPGR-XLRP* mutant dogs. In a previous study affected XLPRA2 dogs received a subretinal injection of an AAV2/5 vector coding for *hRPGRstb*, the expression of which can be detected by IHC, allowing the delineation of the treated area. (**a**,**a′**,**a″**) Cryosections of a *RPGR*-XLRP mutant dogs’ treated retina labelled with DAPI, RPGR, and H3K27me3. Note the presence of the RPGR in the inner segments (IS) and the absence of H3K27me3 accumulation in the retina. (**b**,**b′**,**b″**) Same labellings for a non-treated dog. Remark the absence of the RPGR labelling and the presence of an H3K27me3^h^-positive cells in the ONL (arrow) which is mislocalized. (**c**) Quantification of the H3K27me3^h^-positive cells in the ONL of WT and mutant dogs at different ages and (**d**) in AAV2/5- *hRPGRstb* vector treated (Tx) and non-treated (NT) *RPGR*-XLRP mutant dogs. For area definition, please see the material and method section. (**e**,**e′**) magnification of the H3K27me3^h^-positive cell in (**b**″). Another example of H3K27me3^h^-positive cell which is present in the degenerating ONL is present in (**f**,**f′**).

**Table 1 ijms-22-09331-t001:** Animal models expressing H3K27me3^h^ during retinal degeneration.

Disease Form	Animal Model	Age	H3K27me3^h^
Recessive RP	*Rd1* (mouse)	PN12	+++
	*Rd10* (mouse)	PN20	+++
	*Rho^−/−^* (mouse)	PN30	++
	*Rcd1* (dog)	PN28	++
Dominant RP	*Rho^P23H^* (mouse)	PN15	+++
	*Rho^S334ter^* (rat)	PN12	+++
Ciliopathy	*Fam161a^tm1b/tm1b^* (mouse)	PN30	++
	*BBS10* (mouse)	PN14	+
	*XLRPA2* (dog)	5 weeks	++
		26 weeks	+

The “+” symbol indicates the appreciation of the frequency of H3K27me3^h^-positive cells in one retina slice.

**Table 2 ijms-22-09331-t002:** List of antibodies used in the study.

Antibody	Application	Dilution	Supplier (Catalog #)
Anti-H3k27me3	IHC/WB	1:400/1:2000	Millipore (07-449)
Anti-H3k27me3	IHC	1:1000	Abcam (AB6002)
Anti-H3K9me2	IHC	1:500	Millipore (07-212)
Anti-H3K4me3	IHC	1:500	Millipore (07-473)
Anti-H3, ct	WB	1:5000	Millipore (07-690)
Anti-BMI1	IHC	1:100	USBiological (B2185)
Anti-BMI1	WB	1:2000	Boster immunoleader (PB9133)
Anti-Rhodopsin	IHC/WB	1:1000/1:1000	NeoMarkers (MS-1233-P)
Anti-recoverin	Western blot	1:1000	Chemicon (AB5585)
Anti-cGMP	IHC	1:1500	H.Steinbusch (MHENS)
Anti-CDK4	IHC	1:50	Santa Cruz (SC-601)
Anti-RPGR	IHC	1:100	Sigma (HPA001593)
Anti-EZH2	IHC	1:125	ABGENT (AM1836A)
Anti-EZH2	WB	1:2000	Cell Signaling (3147s)
Anti-GFP	IHC/WB	1:1000/1:1000	Abcam (ab290)
Anti-GAPDH	WB	1:2000	Chemicon (MAB374)
Anti-Glutamine synthase	WB	1:50	Millipore (MAB302)

IHC: immunohistochemistry; WB: Western Blot. #: number.

## Data Availability

All data are contained within the article and Appendix A.

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
