# Peer review of "Enhancer of Zeste Homolog 2 (EZH2) Contributes to Rod Photoreceptor Death Process in Several Forms of Retinal Degeneration and Its Activity Can Serve as a Biomarker for Therapy Efficacy"

_ijms, 2021, doi:10.3390/ijms22179331_

Round 1

Reviewer 1 Report

In this paper you have investigated about an interesting topic. However, a low number of issues need to be addressed. You need to deepen some aspects and update bibliographic references (Porciatti at al. / Romano at al.) Moreover, you need to slightly fix few points in the text.

Author Response

In this paper you have investigated about an interesting topic. However, a low number of issues need to be addressed. You need to deepen some aspects and update bibliographic references (Porciatti at al. / Romano at al.) Moreover, you need to slightly fix few points in the text.

  1. P2X7 receptor antagonism preserves retinal ganglion cells in

   glaucomatous mice

   (doi: 10.1016/j.bcp.2020.114199)

   (https://pubmed.ncbi.nlm.nih.gov/32798466/#)

  1. Modeling Retinal Ganglion Cell Dysfunction in Optic Neuropathies

   (https://pubmed.ncbi.nlm.nih.gov/34198840/#)

   ((Vittorio Porciatti et al. Cells. 2021. Free PMC article Hide details

     Cells )

Response: we have taken into account the remark, added the two references and changed the text in relation to these works. In page 12, line 351: “The present study focused on photoreceptor degeneration, which degenerative processes are different to those documented for the retinal ganglion cells [30, 31]. It would be nonetheless interesting to reveal whether EZH2 pathway or other epigenetic modifications occurs during glaucoma or other models of optic neuropathies [32, 33]. The use of the Oct4 (also known as Pou5f1), Sox2 and Klf4 genes (OSK) to reverse DNA methylation during glaucoma and restore vi-sion[34] strongly suggest that PRC2 may be involved in glaucoma as well.”

Reviewer 2 Report

This is a well-organized study to demonstrate the role of EZH2 and H3K27me3h in the later cell death of photoreceptors. Their observations may provide new treatment target for inherited retinal degeneration.

Only two minor concerns:

  1. In Fig.5, non-treated figure b'' showed only one  H3K27me3h–positive cell presented, which is different from Fig.5d, authors need adjust the figure or other explanations
  2. In Fig.4, no global changes were observed for the H3K27me3h 266 mark in the same area after UNC1999 injection. Authors should discuss the possibility of un-expected results.  Does the injection timing or injection direction and drug half life matter?

Author Response

This is a well-organized study to demonstrate the role of EZH2 and H3K27me3h in the later cell death of photoreceptors. Their observations may provide new treatment target for inherited retinal degeneration. 

Only two minor concerns:

1.In Fig.5, non-treated figure b'' showed only one H3K27me3h–positive cell presented, which is different from Fig.5d, authors need adjust the figure or other explanations

Response: We apologize for not being more precised. The term “area” represents the treated or non-treated area (Fig5d) and are defined in the Material and Method section, and did not correspond to one microscope field (Fig5b). The picture is thus in accordance with the graph. We have adjusted the Figure 5 legend text (page 11, line 290) as follow. “For area definition, please see the material and method section.”

In the material and method (page 15, line 448), you can find: “Cell counting in the RPGR-XLRP mutant dog retina slices: The number of highly H3K27me3-positive cells in the ONL was counted in the AAV2/5-hstbRPGR treated area, versus the non-treated area. For dog Z412, the equivalent of six 20x fields of treated area versus six 20x fields of non-treated area were analyzed; for dog Z459 three 20x fields in the treated area versus seven untreated and for the Z460, 7 20x fields were investigated for each treated and non-treated area.”

2.In Fig.4, no global changes were observed for the H3K27me3h mark in the same area after UNC1999 injection. Authors should discuss the possibility of un-expected results. Does the injection timing or injection direction and drug half life matter?

Response: Page 12, line 328, we completed the discussion section after “The results obtained with the Fam161a knockout mice tend to support this observation when performing multiple injections. However, so far the effect is modest probably due to the drug’s short half-life, a rapid vitreal clearance, and the lack of homogeneous distribution throughout the retina” as follow: “Indeed, the number of H3K27me3-positive cells was not diminished 4 days after the last injection indicating a rapid loss of the injected inhibitor. In Sahaboglu et al[24], the authors estimated the half-life of olaparib to be eight minutes (which is a larger molecule than UNC1999) strongly supporting as well a rapid clearance of the EZH2 inhibitor. Our results also revealed that EZH2 hyperactivity rapidly recovered when the inhibitor is absent. An ocular formulation has to be developed to improve the drug efficiency and distribution”.
